



# Analysis of seasonal climate and streamflow forecasts performance for Mainland Southeast Asia

Ubolya Wanthanaporn[1], Iwan Supit[1], Bert van Hove[1], Ronald W.A. Hutjes[1]

[1]Water System and Global Change Group, Wageningen University, PO Box 47, 6700 AA Wageningen, The Netherlands

5   *Correspondence to*: Ubolya Wanthanaporn (ubolya.wanthanaporn@wur.nl)

**Abstract.** Seasonal forecast is an early warning system that contributes to anticipatory management by providing spatial and temporal information of the near future. This study first examined the skill of ECMWF system 5 (SEAS5) sub–seasonal–to–seasonal (S2S) forecasts over Mainland Southeast Asia (MSEA). We evaluated the SEAS5 skill of temperature and precipitation for 30 years (1985–2014) against two reference model datasets, WFDE5 and APHRODITE, using probabilistic 10   forecast verification skill metrics at grid cells for each month. Then, the SEAS5 data was used to force the Variable Infiltration Capacity (VIC) hydrological model to predict runoff and streamflow. These hydrological results were compared against the WFDE5-driven streamflow reanalysis and observed station data, using the same probabilistic skill statistics. The results show a prediction potential for temperature beyond two months in advance. The skill of precipitation and streamflow forecasting is limited to the first month. Strong seasonal and regional dependence occurs. The model shows high forecast 15   skills during the pre-monsoon (April–May) and post-monsoon (October–November), arguably the period when its usefulness is potentially highest. Conversely, poor skill is observed during the rainy monsoon season (June–August). In eastern and southern MSEA, i.e. in eastern Thailand, Cambodia, Vietnam and Malaysia, considerable skill levels are found. Year–to–year precipitation tercile plots highlight skill in predicting the anomalous seasonal conditions caused by the ENSO. Overall, SEAS5 and derived hydrological forecasts show useful skill that can potentially be used for hydrological and agricultural 20   anticipatory management in this region.

**Keywords:** Mainland Southeast Asia, seasonal forecast, streamflow, ECMWF SEAS5

## 1 Introduction

Mainland Southeast Asia (MSEA) is located in the tropical zone, where the climate variability strongly depends on a complex interaction between the ocean and the atmosphere of the Indian Ocean and the Pacific Ocean. Its climate is 25   modulated by monsoons and amongst others the Inter Tropical Convergence Zone (ITCZ) and tropical cyclones contributing to the rainfall during the wet season. Moreover, the irregular oscillations of sea surface temperature, such as the Indian Ocean Dipole (IOD) and the El Niño Southern Oscillation (ENSO), are responsible for anomalous seasons. The ENSO is now recognized as the most important driver for year–to–year climate variability (Lieberman and Buckley, 2012), especially in the tropics. Abnormal ENSO phases can cause the seasonal climate to become wetter or dryer than usual and consequently



affect the hydrology. Generally, drought conditions in MSEA tend to be associated with El Niño, while severe flood events

are correlated with La Niña (Juneng and Tangang, 2005; Villafuerte and Matsumoto, 2015; Xu et al., 2004).

The recent increasing global mean temperature changes regional temperature and precipitation and affects streamflow

(Gosain et al., 2006; Xu et al., 2010). This becomes a challenge to water resource management regarding climate change

adaptation. Decisions in the past have not always been based on, nor supported by effectual information. Clearly, decision–

making needs more research to accomplish more secure and sustainable water management. Hence, probabilistic forecasts

are necessary to establish a strategy taking uncertainties about future hydrological conditions into account.

Various statistical and empirical streamflow prediction models and methods have been employed to support water

management (Schaake et al., 2007). Hydrological planning typically requires forecasts at various lead times. For instance,

short term forecast is used for day–to–day strategy. Seasonal forecasts produce information on such events as ENSO from a

month to over a year in advance and that has considerable potential for an annually varying management strategy.

Furthermore, sub–seasonal–to–seasonal (S2S) meteorological forecasts have been produced to fill the gap between the

medium–range and seasonal weather forecasts for informing decision–making across sectors. Many recent studies have been

conducted to determine the S2S prediction skill. For example, Amalia et al. (2019) studied the skill of the CFSv2 forecasting

model over Southeast Asia. Seasonal climate model outputs subsequently drive statistical or process-based hydrological

models and extend hydrological variables prediction on seasonal timescales. Many seasonal hydrological forecasts have been

developed using variety of forecasting methods. Examples are drought forecasting in Africa by Trambauer et al. (2015) and

seasonal streamflow forecasts over Europe produced by Arnal et al. (2018). State-of-the-art coupled ocean–atmosphere S2S

forecasting systems, such as the one from the European Centre for Medium–Range Weather Forecasts (ECMWF) are

promising prediction methods that have been evaluated in many studies (e.g., Olaniyan et al. (2018); Yan et al. (2021)).

Although forecasting skills have improved, there are still some limitations. For instance, it is still unclear how well SEAS5

represents the regional climate dynamics. We do not know yet how the climate forecast skill, if any, translates into skilful

streamflow forecasts. It is worth assessing its usefulness for meteorological and inflow forecasting. Moreover, it is important

to understand seasonal anomalies associated with the ENSO phases over MSEA to provide advanced knowledge to enhance

implementation and planning for effective water management. Quantifying the forecasting skill for different regions and

time scales is necessary to identify the model error and enhance the effective uptake of forecast information.

In this present study, we implemented a model-based system, the variable infiltration capacity (VIC) hydrological model and

the ECMWF system 5 to produce seasonal streamflow forecasts. It was applied in MSEA. We addressed the following

research questions: (1) How well does the seasonal forecasting model ECMWF SEAS5 simulate the climatology and

seasonality of the variables (i.e. precipitation and temperature) over MSEA? (2) How well does the streamflow prediction

forced by the seasonal forcing variables perform? (3) How does forecast skill for these indicators at various lead times

compare for each season and in different sub-regions?



Section 2 will give details regarding the study area and available data. The methodology and skill validation are given in Section 3. The performance of SEAS5 forecast skill result is presented in Section 4 and discussed in Section 5. We conclude our study in Section 6.

## 2 Study area and data description

### 2.1 Study area

The study area is MSEA, covering Myanmar, Thailand, Laos, Cambodia, Vietnam, and Peninsular Malaysia (Figure 1). MSEA is a tropical humid climate region (Peel et al., 2007). Within MSEA, the Mekong River Basin is a flood-prone area with rich natural resources and is favourable for agricultural activities. The north and northwest areas are high plateaus and mountain ranges across the border between Myanmar and Thailand to Malaysia. The southwest parts are next to the Indian Ocean and the northeast part is next to the South China Sea. The climate is primarily modulated by two main monsoon periods, the southwest monsoon and the northeast monsoon (Loo et al., 2015). The southwest monsoon from the Indian Ocean forces the wet season over the entire MSEA, starting late May and lasting to August or even longer to September. It takes wet moisture to the northeast mountain range along Myanmar-Thailand during the summer season with the maximum rainfall in July. Meanwhile, coastal Vietnam coincides with the maximum rainfall around October, but a lower amount than the northeast region (Kripalani and Kulkarni, 1998). In contrast, the northeast monsoon from southern China brings dry air and dominates the dry and cold season from around November to March (Loo et al., 2015; Misra and DiNapoli, 2014). It induces the dry season over the northern and north-eastern parts and causes rainfall in the southern part. The most relevant oceanic system to this study area is ENSO, which occasionally causes extreme drought and flood events (Lieberman and Buckley, 2012; Räsänen and Kummu, 2013; Xu et al., 2004).

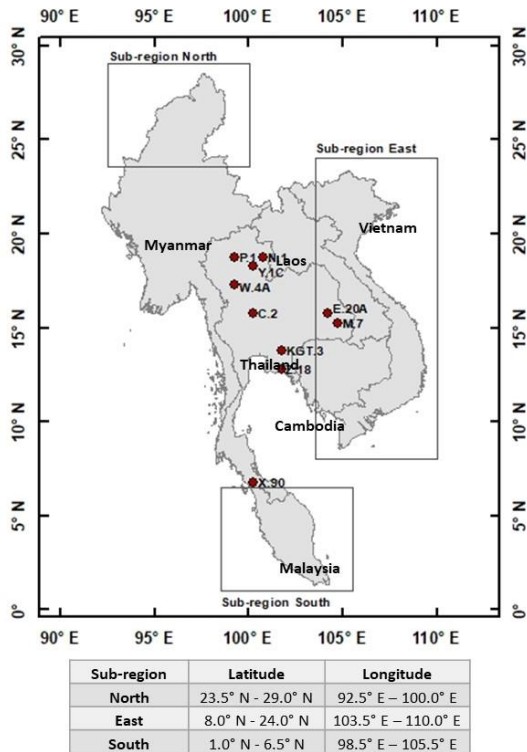

| Sub-region | Latitude | Longitude |
|---|---|---|
| North | 23.5° N - 29.0° N | 92.5° E – 100.0° E |
| East | 8.0° N - 24.0° N | 103.5° E – 110.0° E |
| South | 1.0° N - 6.5° N | 98.5° E – 105.5° E |

**Figure 1 The study area Mainland Southeast Asia (MSEA) and three sub-regions: North, East, and South. The red dots indicate ten streamflow gauging stations from the Royal Irrigation Department (RID), Thailand.**

## 2.2 Data Description

85 The meteorological forcing from the ECMWF ensemble forecast system has been produced since the first system in 1997 (Van Oldenborgh et al., 2005) and has constantly developed to the current version, SEAS5 (Johnson et al., 2019). It is used to produce re-forecasts and real-time forecasts. The SEAS5 re-forecast dataset starting from 1985 to 2014 (30 years) was selected to evaluate meteorological skill in this study. The data is an ensemble of 25 members. The forecast initialization starts on the first day of each month and extends to a lead time of seven months. We performed a bias correction on each

90 ensemble member of precipitation and temperature using the empirical quantile–quantile mapping (qqmap) approach (Themeßl et al., 2011) on daily data against the WFDE5 dataset.

Observational data for the same period is needed for re-forecast (hereafter hindcast) verification. We use both reanalysis datasets and real observed data from the measurement stations as references. We employed the 0.5 degree horizontal resolution gridded reanalysis data, namely the WATCH Forcing Data ERA 5 (WFDE5) (Cucchi et al., 2020) as reference for

95 the meteorological variables. The WFDE5 dataset is the bias-corrected version of the ERA5 reanalysis using the WATCH Forcing Data (WFD) methodology (Cucchi et al., 2020).





In addition, we used the 0.5° gridded daily mean temperature and precipitation datasets from Asian Precipitation–Highly Resolved Observation Data Integration Towards Evaluation of Water Resources (APHRODITE) for validation with the meteorological variables. APHRODITE is long-term precipitation and temperature datasets developed from observation records over Asia (Yatagai et al., 2012). We used the APHRODITE temperature version 1808 and two versions of precipitation data, version 1101 for 1985–1997 and version 1901 for 1998–2014, because neither version covered the entire hindcast period.

Lastly, we obtained the streamflow observed data from the Royal Irrigation Department (RID), Thailand, to compare with the streamflow output from VIC model driven by SEAS5. Ten stations (Figure 1) were selected with complete streamflow data.

## 3 Methodology and skill verification

The study consists of two main parts (Figure 2): (1) a reference simulation of VIC forced by WFDE5 and (2) a seasonal hindcast simulation of VIC forced by SEAS5. We simulated seasonal streamflow forecast with the Variable Infiltration Capacity (VIC) hydrological model WUR version (Droppers et al., 2020) using the SEAS5 data as forcing variables. VIC is a macroscale hydrological model originally developed by (Liang et al., 1994) that simulates full water and energy balance on an individual grid cell. VIC is forced by seven variables: 1) precipitation, 2) minimum temperature, 3) maximum temperature, 4) atmospheric humidity, 5) wind speed, 6) incoming short-wave radiation, and 7) incoming long-wave radiation. For the present study, we ran VIC with a six-hour simulation step. It was first run for the years 1983-1984 with the WFDE5 dataset to spin up the initial state of soil moisture, snow, and river discharge. Subsequently, we used 30 years of SEAS5 and WDFE5 as forcing variables for VIC to generate streamflow. The VIC model was run for the entire hydrological basins in this region to assemble all the sources and outlets of the rivers. Subsequently, the streamflow results were aggregated for the MSEA domain as outlined before. Thus the simulation domain was significantly larger than the analysis domain. The SEAS5-based streamflow results were analysed and compared with the reanalysis streamflow based on the WFDE5 variables. The reanalysis simulation was additionally validated with the gauging observed discharge data from the RID.

Subsequently, the daily hindcast data was aggregated to monthly means. This study examined skill for every 12 targets and seven lead months to achieve a higher temporal resolution instead of analysing target seasons. We further aggregated to three sub-regions (North, East, and South) to examine the different consequences of the ENSO phenomenon.

There is no single verification measure that can capture all forecast qualities, so it is important to use a range of different statistical metrics to assess forecast skills (Murphy, 1993). We calculated three skill statistics (1) correlation coefficient, (2) Ranked Probability Skill Score, and (3) Relative Operating Curve Skill Score. The Spearman's rank correlation coefficient (R) is one of the commonly used measures for forecasting skill analysis. We used the R statistic to verify the correspondence between the observations and the median of the hindcasts. The Ranked Probability Skill Score (RPSS) was applied to



evaluate the hindcast probability to standard references. A zero value means that the hindcast is as good as the climatology
and one means a perfect forecast. Positive or negative scores imply more and less skilful than the climatology, respectively.
The seasonal forecast ability to capture ENSO-associated abnormal rainfall was assessed based on anomalous (seasonal)
years identified from NOAA's Oceanic Nino Index (ONI) (NOAA, 2023) using the Relative Operating Curve Skill Score
(ROCSS). We defined three seasons: March–May (MAM), July–August (JJA) and September–November (SON), and again
the three sub-regions in the North, East and South (Figure 1). A negative ROCSS values mean no skill, zero values imply the
forecast is as good as the climatology, and the positive values indicate an improvement. For the analysis, we used the
following statistical R packages: 'Specs Verification' (Siegert et al., 2020), 'easy Verification' (Bhend et al., 2016),
'visualizeR' (Frías et al., 2018), 'transformeR' (Bedia and Iturbide, 2017a), 'loadeR' (Bedia, 2018) and 'downscaleR' (Bedia
and Iturbide, 2017b).

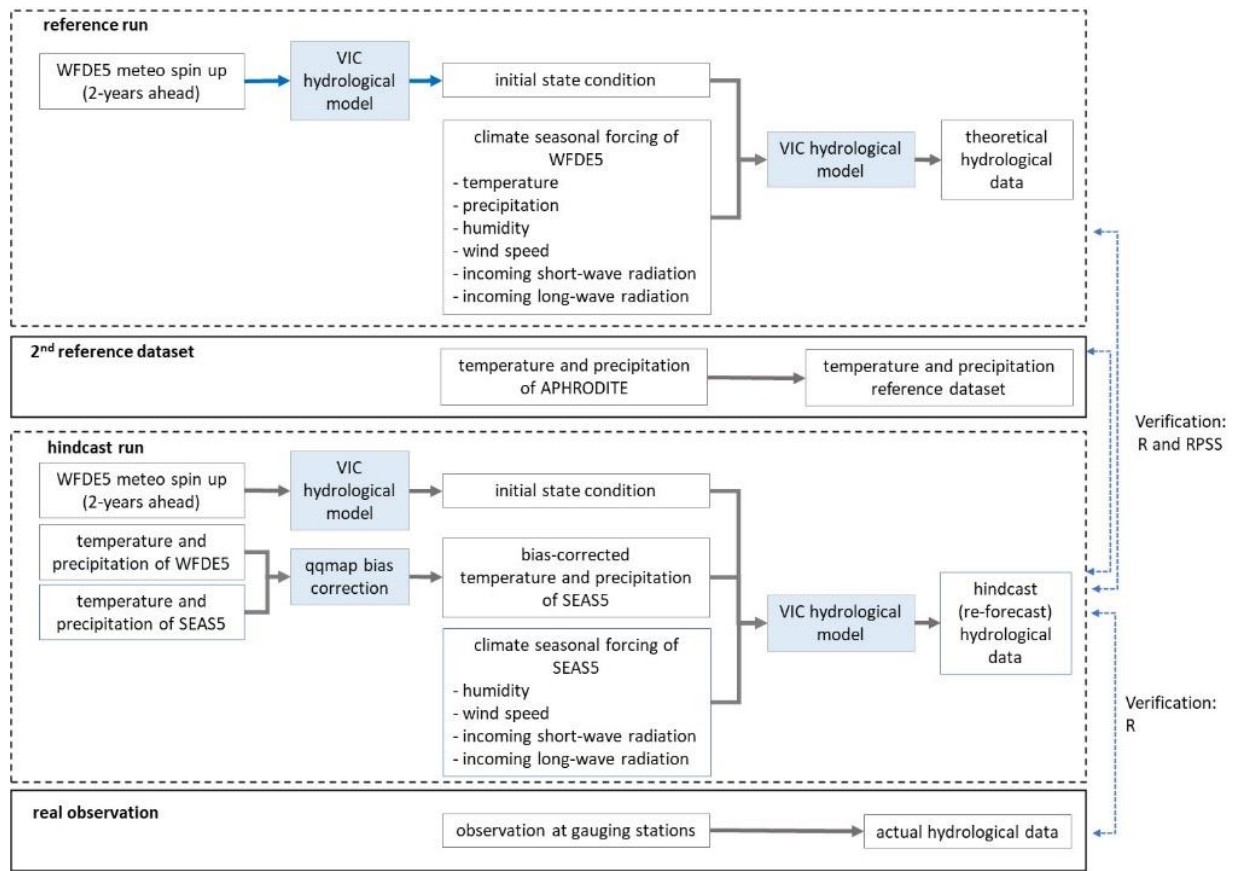

**Figure 2 Set-up of this study. Two dashed boxes present hindcast and reference simulations. Two solid line boxes present obtained datasets: temperature and precipitation from APHRODITE and water discharge from Royal Irrigation Department, Thailand.**





## 4 Results

### 4.2 Spatial–temporal pattern of climatology

In this section, spatial–temporal patterns of temperature and precipitation of SEAS5 hindcast were compared with each 0.5 degree (approximately 55 km) grid cell of WFDE5 and APHRODITE. Maps of the temperature and precipitation skill at 12 targets and seven lead months have been made. Map information has been further aggregated and analysed against time, to facilitate analysis of temporal evolution of skill and its persistence.

### 4.2.1 Near surface temperature

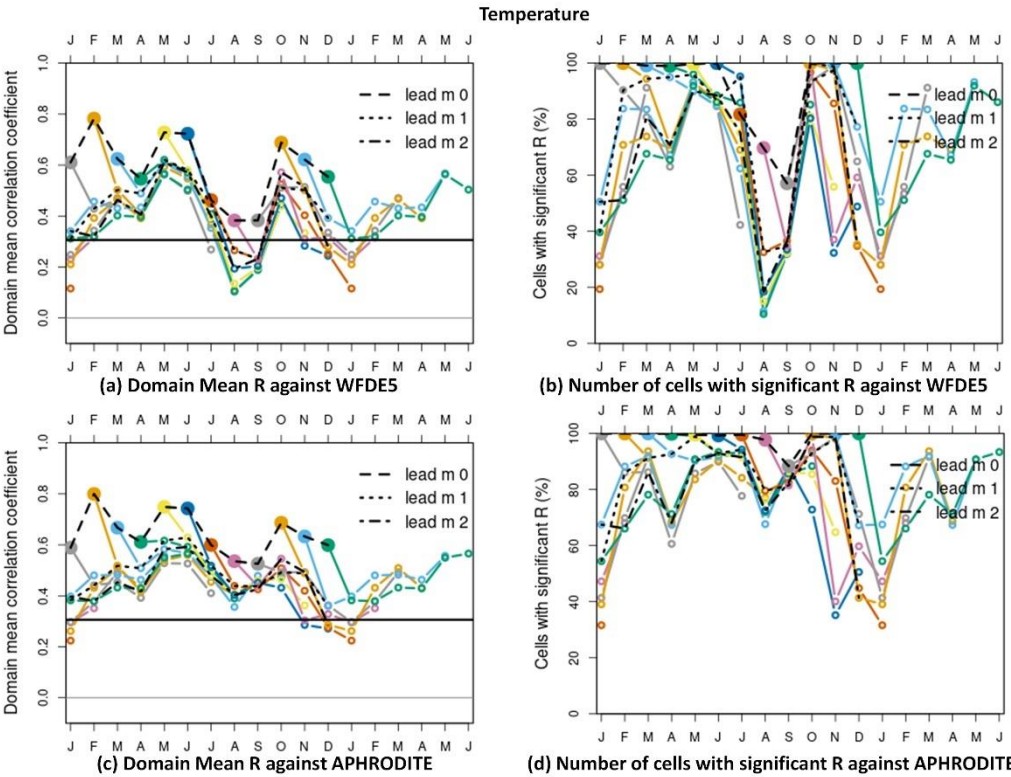


**Figure 3 Spatially aggregated correlation coefficient R (p<0.05) for temperature of SEAS5 hindcast against reference datasets for 1985–2014 over Mainland Southeast Asia at lead 0–2. (a) Mean R against WFDE5; (b) Number of cells with significant R against WFDE5; (c) and (d) are against APHRODITE. Each coloured line follows the skill of a single forecast for its entire seven months. Dashed black lines connects points of the same lead time. The straight line at 0.31 is the significance threshold of mean R for 30**
**years.**

Figure 3 presents the verification result for temperature of SEAS5 hindcast against the WFDE5 and APHRODITE datasets over MSEA for 30 years (1985–2014). The spatial domain mean correlation coefficient shown in Figures 3a and 3c shows an overall trend of decreasing skill with lead time. The results indicate a significant correlation mostly up to two months in advance. The two reference datasets agree on low skill levels during the wet season (July–September), both in mean





correlation and in a fraction of the area with skill. The validation against APHRODITE shows a higher skill magnitude compared with the evaluation against WFDE5, especially during the rainy season. It also can be clearly seen that the number of significant cells during July–September between verification against WFDE5 and APHRODITE is different. The number of significant grids during July–September drops to below 20% at lead-2 against WFDE5, while the significant cells at lead-2 against APHRODITE is about 65%. The hindcast evaluation against WFDE5 reanalysis at $p<0.05$ shows a significant

spatial domain mean correlation coefficient ($R>0.31$) at the initial (lead-0) (Figure 3a and 3c).





**Figure 4 Spatially distributed correlation coefficient R (p<0.05) for temperature of SEAS5 hindcast against reference datasets: (a) WFDE5 and (b) APHRODITE, for 1985–2014 over Mainland Southeast Asia at lead-0.**

Spatial R patterns for the temperature at lead-0 of the verification against WFDE5 and APHRODITE are shown in Figure 4a
and 4b, respectively (other lead times shown in the Supplement). Grids with no significant skills are presented in yellow. The darker red colour grid represents more skill. For various grid boxes, some metrics could not be computed because the observations or hindcasts contained more than one-third of zeros or more than one-sixth equal values: these grids will remain white on the maps. These skill gaps could occur because of e.g. zero precipitation in the dry period (see also Greuell et al. (2018)).

The maps demonstrate poor skill or no statistical significance around the central or central-east part of MSEA, particularly in July–September, which is related to the low mean correlation during the wet season (Figure 3). Even though the evaluation against APHRODITE presents more significant cells in this area compared to the hindcast against WFDE5, the degree of skill is still less than in the other areas. More correlation is found in the central MSEA apart from the wet season. Temperature also shows a strong relationship in all months between SEAS5 hindcast and the reference run in mainland
Malaysia (southern MSEA) and the Myanmar high plateau (northwest MSEA). Though, there is a weaker skill period during July and September, which can be remarkably seen with WFDE5 reanalysis.

### 4.1.2 Precipitation

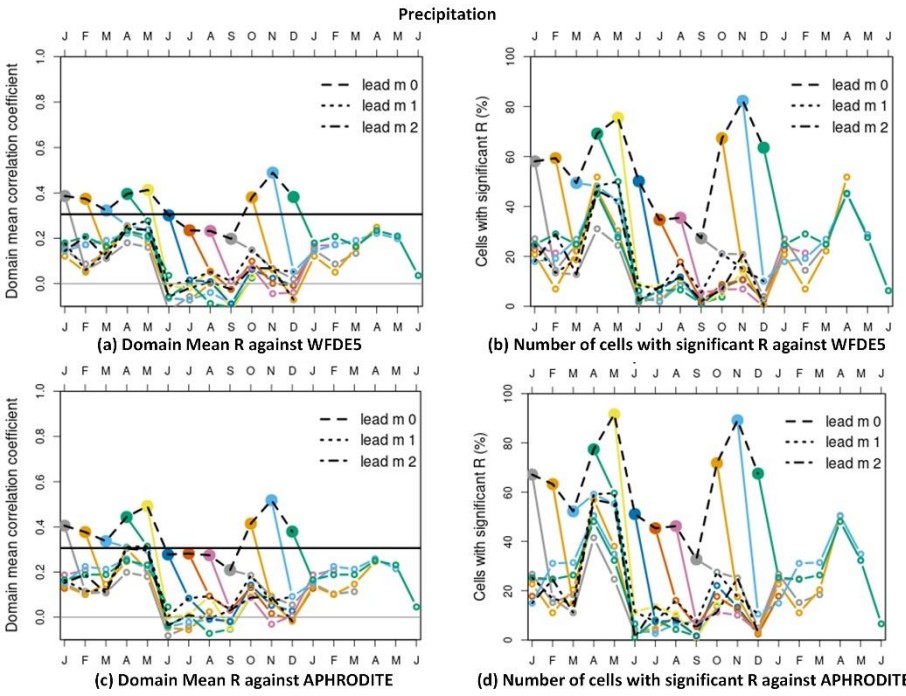





**Figure 5 Spatially aggregated correlation coefficient R (p<0.05) for precipitation of SEAS5 hindcast against reference datasets for 1985–2014 over Mainland Southeast Asia at lead 0–2. (a) Mean R against WFDE5; (b) Number of cells with significant R against WFDE5; (c) and (d) are against APHRODITE. Otherwise as figure 3.**

Figure 5 presents the correlation coefficient R of the SEAS5 precipitation hindcast against the reference datasets. The significant forecasting skill (R> 0.31) is observed only at the initial month (lead-0) over the spatial domain. The evaluation skill results with the two reference datasets are comparable and more similar than for temperature. Verifying with APHRODITE shows a slightly better skill. The skill results are likewise monthly dependent. The R scores are apparently low during the rainy season from June to September, which is the same low skill period for temperature. The skilful target months are the pre- and post-rainy seasons. Beyond the first lead month, the mean R of the precipitation model hindcast over the entire study area decreased with lead time (see more spatial distribution in Supplement).

As the significant precipitation hindcast skill solely appears in lead-0 (Figure 5), we will only show the spatial distribution of R for precipitation against the reference datasets at lead-0 in Figure 6. There is a large area in the northwest and central MSEA (North Thailand and Myanmar) where the statistic cannot be calculated from December to March because of the ill-defined data that occur in the hindcast. The reasons can be that the reference precipitation consists of more than one-third zeros or of more than one-sixth equal values. This can be the consequence of no precipitation in the dry season because these areas are located in the northern highland and the western mountain range blocks the moist air from the Indian Ocean to the central region. The non-computed grid cells cause the lower number of spatial grid cells with significant R during the dry season. The northern and central areas are skilful around the start and end of the rainy season. The southern MSEA displays significant R in the dry season. However, there is no clear significance pattern during the wet season. This might be due to the high rainfall variability in this region as the tropical zone where the maximum precipitation is high and difficult to forecasts.


**Figure 6 Spatially distribution correlation coefficient R (p<0.05) for precipitation of SEAS5 hindcast against reference datasets: (a) WFDE5; and (b) APHRODITE, for 1985–2014 over Mainland Southeast Asia at lead-0.**

## 4.2 Spatial–temporal variation of skill in hydrology

The seasonal streamflow hindcast skill is evaluated for all 12 initialization months and all seven lead months using the 30

years of SEAS5 data as forcing for VIC to generate the hydrological outputs and that will be evaluated with WFDE5-based streamflow outputs. The analysis focuses on runoff (the amount of water that runs within a grid cell via both the surface and groundwater) and river discharge (the amount of water that flows through the river channel between grid cells).





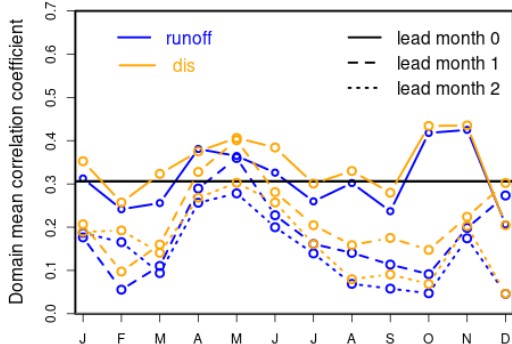

**Figure 7 Spatial domain mean correlation coefficient R (p<0.05) for river discharge (orange line) and runoff (blue line) generated**
**from VIC model driven by SEAS5 hindcast against the reference dataset driven by WFDE5 for 1985–2014 over Mainland**
**Southeast Asia at lead 0–2.**

The SEAS5-based streamflow forecast generally shows little skill. The monthly mean correlation patterns between runoff
and discharge are almost similar (Figure 7), but the prediction is slightly better for discharge than for runoff. The difference
can be explained by the runoff infiltrates in a single grid cell, while discharge accumulates the river flow from upstream to
downstream that aggregates the skill through some time intervals. The difference between discharge and runoff is clearly
visible in the spatial correlation coefficient map (Figure 8 and Figure 9). For example, the forecasting skill for discharge in
March observed in the river channels is higher than the skill in discharge and runoff in the surroundings for the same month
(Figure 8). There are some exceptions, e.g. in target months February and December, where the river channels in the central
area show less skill than the surroundings. The occasional difference is perhaps because the discharge predictability comes
later in response to rainfall upstream. The significant streamflow correlation at each initiation month and lead time
correspond to the forecasting skill of meteorological variables. Nevertheless, the skill level of discharge and runoff hindcasts
is lower compared to the temperature and precipitation skilful. The significant R is observed at the initialization months and
almost no skill after lead-0. Considering lead-0, the discharge skill is small during December–March and July–September
(Figure 7). This may be due to the influence of the monsoon period together with ENSO. Although the line graph of the
entire MSEA shows a low correlation during December–February, the spatial map shows Malaysia and Vietnam and
Cambodia give a significant skill in this dry season (Figure 8). In contrast, the central MSEA exhibits highly skilled during
the wet season (April–June) but shows low skill in the dry season. This result is similar to the precipitation skill pattern
showing a strong relationship between precipitation and hydrology.



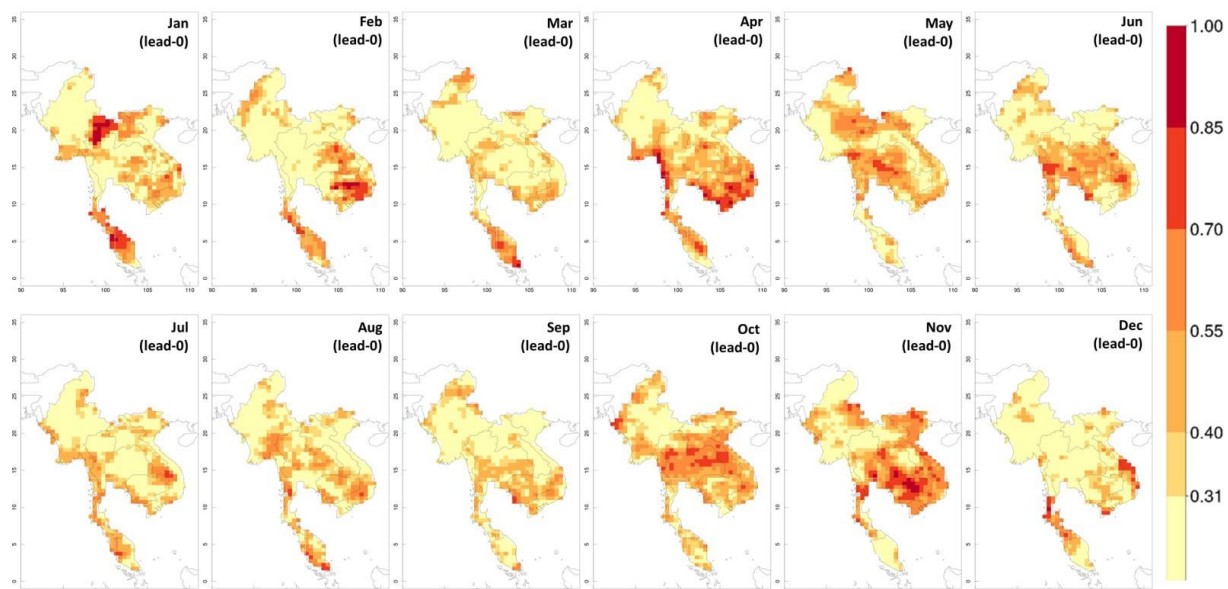

**Figure 8 Spatially distributed correlation coefficient R (p< 0.05) for water runoff generated from VIC model driven by SEAS5 hindcast against the reference dataset driven by WFDE5 for 1985–2014 over Mainland Southeast Asia at lead-0.**

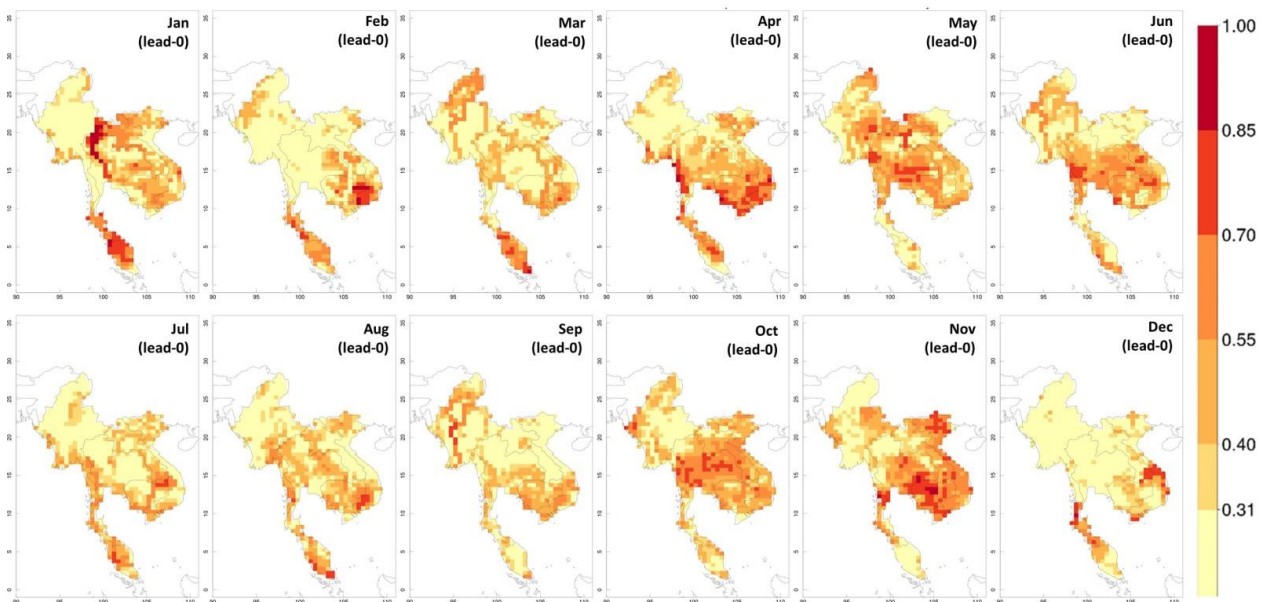

**Figure 9 Spatially distributed correlation coefficient R (p<0.05) for water discharge generated from VIC model driven by SEAS5 hindcast against the reference dataset driven by WFDE5 for 1985–2014 over Mainland Southeast Asia at lead-0.**

## 4.3 Other skill metric (RPSS)

So far, we have only calculated the correlation coefficient R approach to test the seasonal hindcast skill. We also applied the RPSS evaluation test in addition to the R statistic. The RPSS evaluation also shows the temporal dependencies in this region.





We present here the target May up to two lead times (Figure 10 and Figure 11), which is the month with the highest skill. The positive RPSS is presented in light green to blue for precipitation and orange to red for temperature, referring to significant cells where the forecast is better than the climatology and potentially useful, while the yellow colour refers to insignificant RPSS levels. Overall, it can be seen that the seasonal hindcast RPSS skill decreases with longer lead times. The spatial gridded RPSS for seasonal temperature hindcast resemble the R results, the positive RPSS results are mostly found in the central areas of MSEA (Figure 10a). The RPSS values advocate seasonal temperature forecast effectiveness. Even though the model seasonal precipitation, discharge and runoff hindcasts rarely produce RPSS skill, the high RPSS regions are similar to and slightly more extensive than the high R regions. This agreement gives more robust indication of which regions and what time periods the seasonal hindcast is skilful. The precipitation and streamflow results are highly correlated in terms of skill patterns. This emphasized the strong relationship between precipitation and the hydrological system. In other words, the runoff and discharge forecasting skills predominantly depend on the precipitation forecasting skill and apparently less so on initialization.

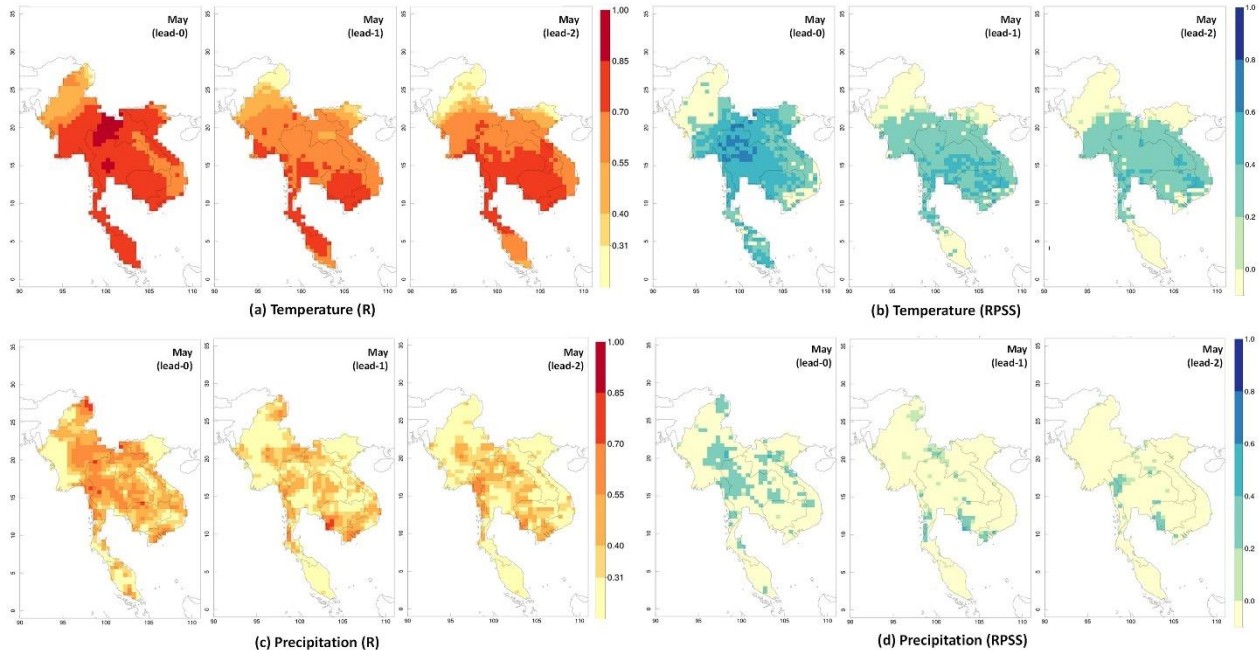

**Figure 10 The skill of near surface temperature by (a) mean correlation coefficient R (p<0.05); and (b) ranked probability skill score RPSS (p<0.05) of SEAS5 hindcast against observation from WFDE5 for 1985–2014 over Mainland Southeast Asia at lead 0– 2. (c)-(d) are the same for precipitation.**





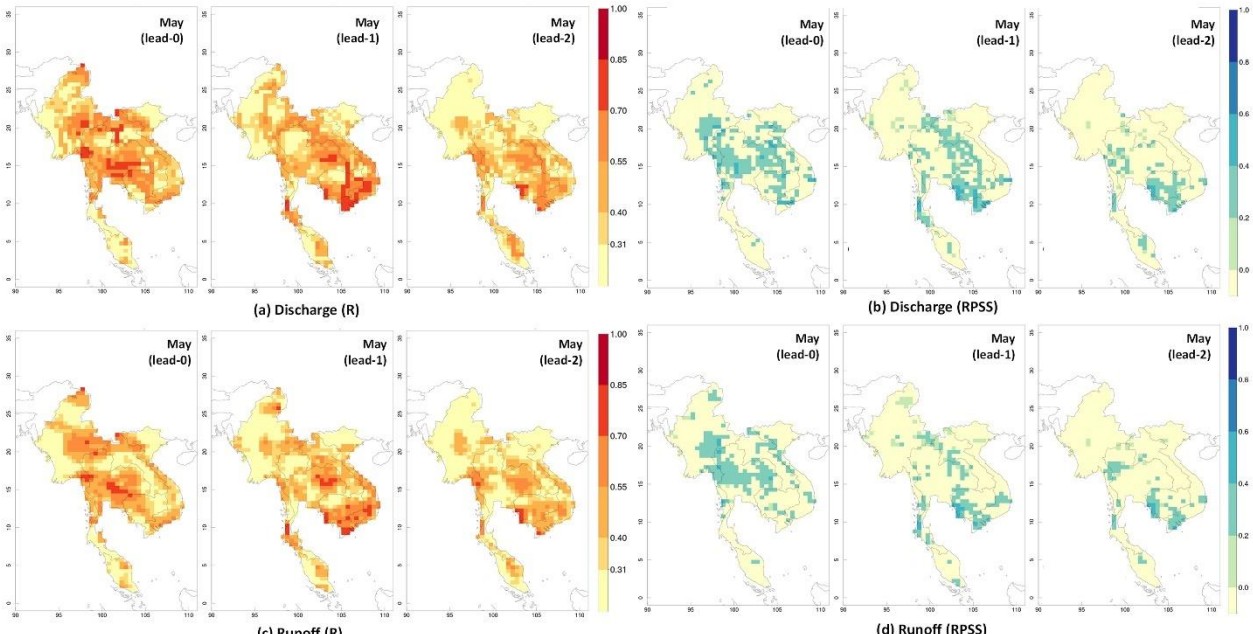

**Figure 11 The skill of river discharge generated from VIC hydrological model driven by SEAS5 hindcast against the reference dataset driven by WFDE5. (a) mean correlation coefficient R (p<0.05); and (b) ranked probability skill score RPSS (p<0.05) for 1985–2014 over Mainland Southeast Asia at lead 0–2. (c)-(d) are for runoff.**

### 4.4 Comparison of theoretical skill and actual skill of discharge

All the previous results in this study were analysed against the generated reference model simulation. To better assess the
value of the hindcast, skill verification against observed data is required. The terms 'theoretical and actual skill' was introduced for skill validating with the reanalysis method and real observational data, respectively and commonly used in many studies (Greuell and Hutjes, 2022; Van Dijk et al., 2013). Details of the streamflow gauging stations are presented in Supplement Table S1.

Figure 12 shows the probability R values of 12 initial months in seven lead times against the observational discharge data
(actual skill, left column) and against the VIC results forced by WFDE5 (theoretical skill, right column) at 6 gauging stations (the other station results can be found in the Supplement). Significant skill levels are displayed in yellow to red colour, and the remaining grey is an insignificant skill. The notable outcome is the similar skill pattern between theoretical and actual skills. Theoretical skill generally presents slightly higher skill degrees than actual skill.

Station P.1 and N.1 are located in the north of Thailand (central MSEA) and show a low number of significant R results with
unclear skill patterns. Even though some skills are found within these stations, it might be a chance that skill is attributed through the analysis process. On the other hand, the skills of other gauging stations show a more consistent skill pattern. Station C.2 (central Thailand), M.7 (east Thailand) and KTG.3 (southeast Thailand) reveal forecasting skill when the target month is May and some skills for April and June and a persistence of skill through a large number of lead months. Although





the actual skill degree is lower than the theoretical skill, the results of both assessments show a lot of agreement. Hence, the

reanalysis can be considered as a reliable method to evaluate the seasonal forecast skill over this area.

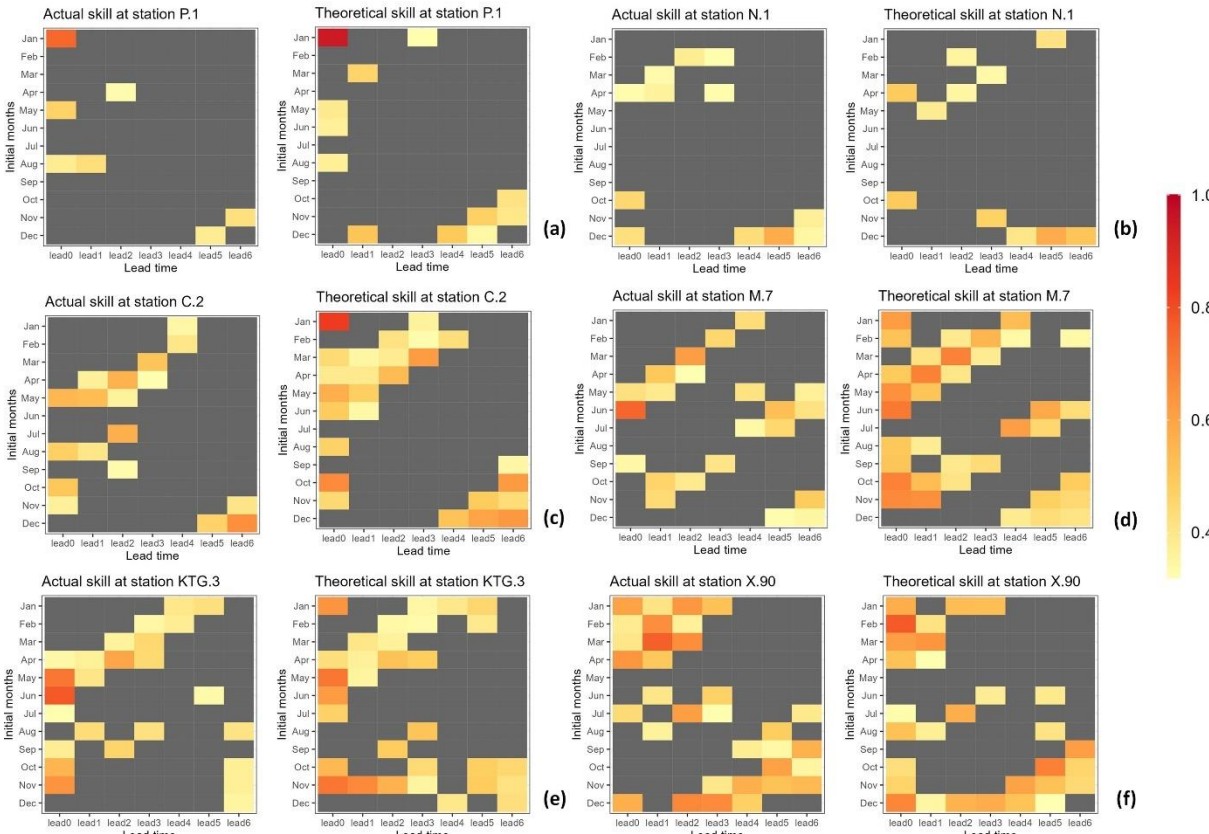

**Figure 12 Comparison between river discharge from gauging stations, known as actual skill (left column), and river discharge generated from VIC model driven by SEAS5 hindcast, known as theoretical skill (right column), using the correlation coefficient R (p<0.05). (a) station P.1; (b) station N.1; (c) station C.2; (d) station M.7; (e) station KTG.3; and (f) station X.90.**

**4.5 Prediction of anomalous years**

Infrequent events such as severe floods or extreme droughts are associated with ENSO events. We tested whether it is possible to capture anomalous weather and discharge with seasonal forecasts. Figure 13-15 demonstrates the year–to–year precipitation hindcast probability in tercile plots from 1985–2014 for the whole study area in three seasons. These tercile plots show the performance of a forecast system at different time scales and compare the different thresholds of hindcast

probabilities (light to dark red blocks) to the observations (white circles). The darker the colour, the better the more certain the forecast is. We analysed the three sub-regions in the North, East and South and for three seasons, defined as pre-monsoon phase (MAM), the rainy season (JJA) and the transition phase (SON). MAM is the dry and hottest period of the year. The rainy season during JJA is dominated by warm moist air from the southwest monsoon. SON is the transition phase from the southeast to northeast monsoon periods. The ROCSS for above normal, normal and below normal on the right of



the plot indicates the forecasting skill in comparison to the reference dataset. The hindcast predictabilities for above and below average precipitation hindcasts are more skilful (higher ROCSS) than in normal conditions. The highest skill is found at lead-0 and reduces at longer lead times.

Even though the SEAS5 hindcast is not able to capture all the anomalous dry and wet years over the 30 years, the hindcast exhibits a remarkable possibility of capturing the extreme coincidences during the pre-monsoon season (MAM).

Furthermore, we detected the predictability of the wet (La Niña) and dry (El Niño) phases during these months. For instance, a strong probability is found for the above-normal due to the La Niña in 1999, 2000.

The observed precipitation terciles in the JJA period that are associated with the El Niño phases are predominantly above normal or normal, while the La Niña phases are below normal (Figure 14). The hindcast hardly detects abnormal rainfall during the JJA monsoon period after the first lead month. The exception is in the South, where the ENSO effect is opposite

to the other sub-regions and anomalous rainfall due to La Niña can be identified by the SEAS5 hindcast. Significant ROCSS values are observed at the upper and lower terciles in the South during the JJA months, except the lower tercile at lead-1. This emphasizes the predictability for the South sub-region.

Among the three selected seasons, the SON period is the most affected by the ENSO events. The hindcast can predict anomalous years in MSEA, except in the South where significant ROCSS values are rarely observed. In the East, we clearly

see that El Niño mainly occurs in the below tercile and La Niña mainly in the above tercile, this shows that precipitation is more predictable and consequently results in higher ROCSS values. (Figure 15). There is an unclear precipitation trend in the North and South as El Niño events occur in both upper and lower terciles and the same for La Niña events, which explains the forecasting difficulty in this transition phase.

Overall, more forecasting skill is observed during the MAM than in the JJA and SON periods, except in the South sub-

region, where the hindcast performs better during the JJA months. These results are consistent with the probabilistic R and RPSS presented before.



**Figure 13 Year–to–year precipitation hindcast probabilities in tercile plots for March–May (MAM) by SEAS5 from 1985 to 2014 at lead 0–2 over (a) Mainland Southeast Asia; (b) Sub-region North; (c) Sub-region East; (d) Sub-region South. White circles indicate observed precipitation (WFDE5) occurred in that particular tercile, black stars indicate El Niño years, black squares indicate La Niña years, and asterisks indicate significant ROCSS at p<0.05.**

Figure 14 As figure 13 but then for precipitation hindcast probabilities in June–August (JJA)



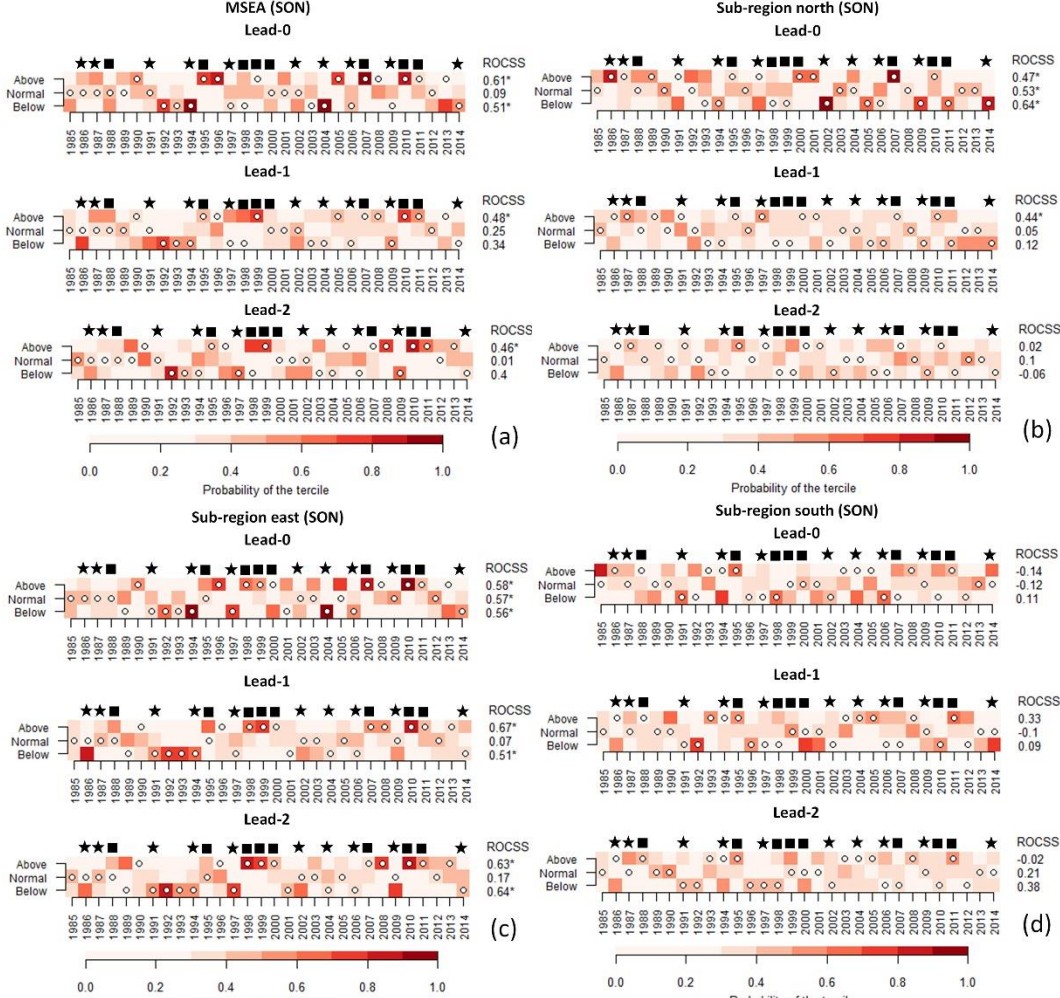

**Figure 15 As figure 13 but then for precipitation hindcast probabilities for September–November (SON).**

## 5 Discussion

### 5.1 Performance of the SEAS forecasting skill

The forecasting skill in MSEA varies by the sub-region and target month. Skilful areas calculated by R and RPSS statistical methods are similar in terms of both spatial and temporal patterns. Significant temperature forecasting skill is shown up to

two months in advance. Even though the temperature generally shows high skill, there are some periods and areas where there is hardly any skill. This could be a challenge for forecasting because the temperature can shift towards higher temperature and higher peak temperature due to climate change (Supharatid et al., 2021; Villafuerte and Matsumoto, 2015). At the same time, for monthly precipitation, there is no forecasting skill after the first lead month (lead-0). This is not a surprising result as other research also found that the SEAS5 forecasts temperature better than precipitation, e.g. in Australia





(Wang et al., 2019). MSEA is located in the tropical zone where the high rainfall variability (i.e. intensity and frequency) and extreme convective peak obstructs forecasting. It indicates that in the tropical region, predicting precipitation is even a bigger issue.

The climate variability in MSEA is dominated by two main monsoon periods, northeast monsoon and southwest monsoon. These monsoon periods are considered as the source of the forecast. For instance, accurate monsoon onset prediction can
improve seasonal forecasting skill. Yan et al. (2021) have studied the predictability of the northeast monsoon onset with the SEAS5 data, and conclude that the monsoon onset date can be predicted ten days in advance. This accuracy definitely contributes to the capability to predict seasonal rainfall over short lead times. Chevuturi et al. (2021) exposed that the southwest monsoon onset could be forecasted one month in advance according to SEAS5. Wang et al. (2021) evaluated the seasonal forecast results for the East Asian summer monsoon implementing Beijing Climate Center Climate System Model
and also found that the forecast skill dropped dramatically from lead-0 to lead-1 and then fluctuated beyond lead time. These related studies highlight the importance of how the monsoon season affects seasonal weather and what can or cannot be predicted.

The seasonal hydrological hindcast shows a lack of skill beyond lead-0. The predictive streamflow skill in the studied region decreases with the increase in lead times. Similar results have been reported to other hydrological studies (Greuell and
Hutjes, 2022; Lucatero et al., 2018; Petry et al., 2021). The research in other areas, such as Denmark, showed limited streamflow skill beyond a one month lead time (Lucatero et al., 2018) and the same in South America where the peak streamflow could be predicted one month in advance (Petry et al., 2021). The streamflow generation processes mainly come from the precipitation (Schmitt Quedi and Mainardi Fan, 2020), the streamflow skill hence presents similar patterns to the precipitation skill although at slightly lower levels. It could be the complexity of the streamflow itself that reduces skill.
Besides, the uncertainties of the hydrological model would minimize skill. Despite this, the similarity suggests that where and when the seasonal meteorological forecasts are skilful, it adds value to streamflow forecast (Jackson-Blake et al., 2022; Lucatero et al., 2018).

Meteorological and hydrological forecasting skills in MSEA do not predominantly depend on the lead time but more on the target month. The research by Yuan (2016) also found that the seasonal hydrological forecasting of the Yellow River basin
has a strong seasonal dependency. Our study demonstrates that the highest skill occurs for target May, whereas target September is the less skilful month. Apparently, because the monsoon influences precipitation predictability differently for each season which propagates to similar results in streamflow. Research by Yang et al. (2008) studied the southwest monsoon forecast using the NCEP Climate Forecast System model and found that there was forecasting skill at the start of the monsoon (May–June), but the monsoon retreat (September) showed limited forecasting skill.

**5.2 Trust in skill verification method**

We applied the WFDE5 reanalysis dataset to establish the initial state conditions for VIC, to evaluate the hindcasted meteorological variables and to simulate seasonal streamflow reference data. We decided to use this reanalysis approach



because no actual grid observation dataset is available for the regional scale. Nonetheless, the SEAS5 and WFDE5 datasets are created with almost the same atmospheric model (IFS cycle version 43r1 for SEAS5 and version 41r2 for WFDE5)
(Cucchi et al., 2020; Johnson et al., 2019) and consequently the results of these systems are not independent, and comparison could be biased. Therefore, we compared WFDE5 also with APHRODITE, which is an alternative and more independent reference dataset. The almost identical results of these two models build trust in our study using the WFDE5 reanalysis method.

Moreover, the comparable streamflow results between the reanalysis approach and the actual observational data enhances
confidence in using the reanalysis approach. However, there is a difference in terms of skill degree, the theoretical skill is higher than the actual skill. Errors in the hydrological measurements, notoriously so for flooding events could reduce data accuracy and therefore lead to lower actual skill. We nevertheless expect that the measurement errors are not as large as the model errors and only have a small effect on the actual skill. The hindcast and the model reanalysis initial conditions are both generated by WFDE5, which differ from the real streamflow and could cause dissimilarity (Greuell et al., 2019). VIC
performs well in many regional and hydrological contexts although there are still some sensitivities in model prediction (Nijssen et al., 2001; Yan et al., 2015; Yun et al., 2021). Despite the skill level dissimilarity, the reanalysis is a potential method for skill analysis.

To reassure the competence of reanalysis method, more observations covering this study area for a prolonged period are needed to assess the actual skill better and contribute to a better understanding of the seasonal forecasting skill. Still, this
obviously would entail a long-term investment as it takes several decades before time series are long enough to allow such analysis. Therefore, comparing different reanalysis models method is more time efficient.

## 5.3 Aggregated seasonal skill of anomalous years

We used the ROCSS to evaluate the performance of the tercile forecasts over seasons (MAM, JJA and SON) rather than over individual months. The assessment over the MSEA finds obvious skill patterns with the upper and lower terciles exhibiting
more skill than the climatology. The ROCSS values for the three seasons and three sub-regions underline the dependency of lead time hindcast skill on seasons and regions. The ROCSS score is higher than those R and RPSS because the seasonal aggregates increase skill. The skill value decreases with a longer lead time, same as the R and RPSS results, though in many areas remaining significant to lead-2 for the variable analysed. The decreasing skill at higher lead times was also seen in other studies, such as Ogutu et al. (2017) and Chen et al. (2019).

The ENSO phenomenon forces precipitation anomalies differently among sub-regions and seasons. In MSEA, the warm phase (El Niño) tends to be below the normal precipitation and the cold phase (La Niña) tends to be above normal (Kirtphaiboon et al., 2014; Räsänen and Kummu, 2013; Sein et al., 2015). This tendency is apparent in the MAM and OND seasons but opposite in the JJA months. The SEAS5 can capture the strong ENSO anomalous incidences (e.g. El Niño in 1997/1998 and La Niña 1999, 2000, 2001) yet still miss a few of the other incidences. It is because rainfall in this region is
complex and is influenced by many processes apart from ENSO.





The MAM period exposes the strongest probability terciles for the entire study area and the ENSO circulation could be predicted well by the forecast system. In comparison, the JJA and SON periods show less clear evidence to predict ENSO-correlated events. This might be because high rainfall variability hampers the prediction. However, the number of ENSO events that dominated the MAM pre-monsoon season during the past 30 years is limited compared with more ENSO events during the JJA and OND seasons. This possibly cannot represent the skill well; therefore, studying a longer period could assist this understanding.

### 5.4 Implication and recommendation

Seasonal forecast information may be beneficial for decision-making. This research focussed on the regional scale to examine the potential use of SEAS5 and VIC as an early warning system for rainfall and river discharge anomalies over MSEA. So, the detailed seasonal forecasting skill found by this study could be applicable for many specific purposes. To elaborate, the seasonal forecast is important not only for the agricultural sector but it is also important to the water management, energy, and industrial sectors (Block, 2011; Everingham et al., 2002). For example, the prediction of discharge amount is inherent to the capability of hydropower production. The seasonal streamflow forecast can also be used to improve anticipatory flood management and reservoir operation management (Kompor et al., 2020). According to our study, the SEAS5/VIC-based meteorological and streamflow forecasts are skilful for one month in advance. But meteorology-related planning and water management in this area still faces forecasting difficulties for advance planning beyond one month. However, the SEAS5 is valuable for short-term planning. For example, the skilful first month forecast for the early and late phases of the monsoon from SEAS5 can effectively be used for crop management, such as planning sowing and harvest date, especially for rice cultivation as this is the main crop in this region. The low forecasting skill during the wet period may be less relevant for farmers' decisions since during this period water limitation is generally not an issue.

The current study shows the scientific perspectives, which is the first step in developing an early warning system. Consulting with stakeholders will be the next step for an effective implementation of such a system. The agricultural sector and particularly farmers, for instance, should be consulted. We must understand their needs and limitations for effective implementation of adaptation measures using the results of this study and building trust in seasonal forecasting (Ebhuoma, 2020; Nyadzi et al., 2019). In addition, other seasonal forecasting systems and hydrological models should be studied and compared with this study, as they may exhibit better or complementary skill in this region. Our findings demonstrate the forecast that could provide a possible link to regional level end users. As such, it provides a good basis for applying seasonal forecasts, additional work could be done by adding information to support the possible use of the seasonal forecast.

### 6 Conclusions

This research evaluates the potential use of the ensemble seasonal forecast model, ECMWF System 5 (SEAS5), for Mainland Southeast Asia. The SEAS5 possesses skills to forecast meteorological and hydrological systems. The seasonal

temperature hindcast verification using R and RPSS shows significant skill up to two months in advance, except the target August and September that only show significant skill at lead-0. The seasonal precipitation hindcast demonstrates an almost similar spatial skill as the temperature but a lower skill degree, which a considerable skill is in the initial month (lead-0). The
temperature and precipitation forecasting skills depend on the target month and location but barely on lead time. Both temperature and precipitation skill levels certainly depend on the target month. April–May and October–November are the most predictable skills periods of the year. During the wet season itself (July–September) SEAS5 is hardly able to predict seasonal weather in the MSEA area. This is caused by the high number of rainy days in combination with a high variability of rainfall intensity during the monsoon periods and ENSO. Different sub-regions also respond to the forecast differently.
We find a large area in MSEA where the forecasting is skilful, including the middle and eastern parts (East Thailand, Cambodia, Laos and Vietnam) and the southern part (Malaysia). The forecasting skill for meteorological forcing variables influences the hydrological streamflow forecast skill; thus, both monthly discharge and runoff hindcasts show comparable patterns to the monthly precipitation hindcast skill with a lower skill degree. Evaluation of year–to–year seasonal precipitation hindcast over the entire period presents a good skill for detecting anomalous years in both El Niño and La Niña
occurrences, specifically during the MAM period. This indicates the potential use of the SEAS5 for impact studies.

According to this study, even though the overall meteorological and hydrological forecasting skill for MSEA beyond one month is limited, this information is valuable. The forecast could be utilized as an input for early warning system for many sectors. Especially the highly skilful pre-monsoon season could be helpful to farmer decisions such as deciding crop type and sowing date. Further research will investigate the influence of seasonal forecasting products such as the meteorological
forcing and the streamflow in crop model implementation. The results of this study could already support a first step to come to potential anticipatory hydrological management in MSEA.

**Author contributions**

Ubolya Wanthanaporn generated and analysed the results. Iwan Supit, Bert van Hove and Ronald W.A. Hutjes provided overall oversight and guidance. Ubolya Wanthanaporn prepared the manuscript with contributions from all co-authors.

**Competing interests**

The authors declare that they have no conflict of interest.

**Acknowledgement**

This research is supported by the Royal Thai Government Scholarship, which was conducted as part of the doctoral research project of the first author. The authors thank the ECMWFSEAS5 research team for providing the forecasting product



available for this work. We also thank different modelling and observation centres for providing the ERA5 and APHRODITE data. We extend our sincere gratitude to the Royal Irrigation Department of Thailand (RID) for providing observational streamflow data.



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

hydrological extremes in the Lancang-Mekong River Basin? Science of The Total Environment, 785: 147322.