# Peer review of "Ranked Probability Skill Score (RPSS)"

_Hydrology and Earth System Sciences, 2023_

## Author Comment (AC1)

Dear reviewer,

On behalf of my co-authors, we appreciate the time you have taken in our paper and the valuable suggestions that are very helpful to enhance our manuscript. You will find below the answer according to the reviewers' comments.

**Reviewer's general comments:**

The main goal of the manuscript is to evaluate the ability of the forecasting model ECMWF SEAS5 to simulate the climatology of precipitation and temperature over Malaysia and the accuracy of streamflow predictions, at various lead times, forced by the seasonal atmospheric model for different seasons and sub-regions.

Because of the relevance for water managers, as reinforced in the introduction sections it is clear that the manuscript is of general interest for HESS´s readers.

There are, however, several points in the manuscript that I believe that need to be considered by the authors in order to improve the paper interest.

In my view the manuscript needs to be reduced and focus in analyzing what is relevant in a tropical country, which is rainfall rather than discharge. Then, discussion can concentrate on what matters, which is river discharges. The current version is too long, which makes difficult to read.

This is true that we focus on the hydrological forecast; however, meteorological and hydrology are connected. Moreover, much of the agriculture in the region (e.g. rice) heavily depends on irrigation, not only on rainfall and hydropower is growing sector in all MSEA countries. Both temperature and rainfall are the key components of hydrological system. The VIC model is driven by weather conditions to generate runoff and discharge. Therefore, meteorological forecasts are important to determine the relationship relation or consequence in hydrology part. It is necessary to analyze temperature and rainfall forecasts to provide a more complete hydrological forecast and to get some idea of the propagation of skill through the modelling chain. However, we will improve some figures or leave to supplement. For example, the skill between WFDE and APHRODITE that are similar.

Considering the current limitations of climate model for accurately predicting many month in advance, I wander why the statistics chosen are mainly numerical (thus is, correlation) rather than using categorical indexes of performance.

Thank you for the comment. In our study, we combined both numerical (correlation coefficient and RPSS) and categorical (ROCSS). We first used correlation coefficient and RPSS for general skill assessment and subsequently, we used ROCSS for tercile anomalous year assessment.

**Reviewer's specific comments:**

Lines 30-35. While I understand the advantage for water managers of using probabilistic forecast at intra-seasonal scales, this paragraph regarding climate change might be out of context because the adverse impacts of global climate change increase along decadal time-scales, while the focus of the study are the forecast for lead times of ~ 30 days.

Thank you for the comment. We put our research in the context of climate change because we would like to state the importance of seasonal forecast system. The increasing climate variability and weather extremes, as a result of changing climate, will impact the hydrological system and other sectors. Therefore, there is an increasing need for seasonal forecast as an early warning system in support of management. We will reformulate this paragraph of our manuscript accordingly, possibly by adding something like "Hence, probabilistic forecasts are necessary as the early warning system to establish a strategy taking uncertainties about future hydrological conditions into account."

Data description item: the manuscript has so many acronyms that makes it hard to remember while reading. My suggestion is to include a table in the data description section indicating time and space resolution of each data set.

This is a good suggestion. We will include a similar table with data descriptions with acronyms (you will find it below) in the supplement.

Table S2 Data description. Please note, that all data sets have the same native resolution (0.5°)

| Data | Version | Acronym | Time period |
|---|---|---|---|
| WATCH Forcing Data | ERA5 | WFDE5 | 1983 - 2014 |
| ECMWF ensemble forecast | System 5 | SEAS5 | 1985 - 2014 |
| Temperature APHRODITE | 1808 | | 1985 - 2014 |
| Precipitation APHRODITE | 1101 | | 1985 - 1997 |
| Precipitation APHRODITE | 1901 | | 1998 - 2014 |

90-95. I am not very familiar with the WFDE5 dataset which was used as a reference data. In my experience, what is relevant in tropical areas for hydrological forecast is to remove the errors in rainfall since precipitation has the greatest impact on discharges. How the WFDE5 dataset compares with rainfall estimations derived, for instance form satellite products such as IMERG, etc. I´ve seen validations were performed against the APHRODITE database, which is based in station data, but apparently not for the whole hindcast period.

WFDE5 is ERA5 reanalysis data, bias-adjusted against CRU observed data on temperature and radiation, and CRU or GPCC observed data on rainfall and no of wet days. It is often used for impact studies, including hydrological and agricultural analyses. The WFDE5 itself has been evaluated against meteorological observation and represents good realistic global hydrological

system. It provides a number of output parameters that are required to run the VIC model in this study. Moreover, the range of available data is long and covers the study period of this study. APHRODITE is the dataset specifically for Asian area, so we used it to validate the WFDE5. However, there are still limitations of using APHRODITE, such as the lack of completeness needed for this study period and not all parameters provided to run the hydrological model are available.

130-135. If I understood correctly, the experiments were performed only for ENSO years. How many events were considered? It is not clear for me whether a single experiment (thus is, a single forecast) for each season was carried out; or several runs for each season were for different initial conditions throughout each season was used to calculate the statistics of performance.

First, we want to stress that all simulations were continuous for **all** the available hindcast years, they were **not** confined to ENSO years only. They were used for monthly skill assessment of figures 3-12.

In additions we analysed skill at seasonally aggregated time scales (MAM, JJA and SON), again continuous for **all** hindcast years, in figures 13-15, but highlighting years that can be classified as positive or negative ENSO phases. This because ENSO is an important driver of climate variability in the MSEA region. We analyzed the ENSO using the R packages to evaluate probabilistic tercile and ROCSS as mentioned in the manuscript. The evaluations were run separately for each season throughout 30 years. As the results shown in figure 13-15, each season will be run for the entire 30 years and show the probability tercile of each year. The ENSO events (black stars or black squares) are plotted on the years that the ENSO occurred during that season.

We will try to better explain in our methods section

145, item 4.2.1 Near surface temperature. The authors need to justify all the statistics involving temperature. While I do understand the hydrological implications of the accuracy of temperature forecasts in a temperate country because it has to do with melting, in my view it is of low interest in a tropical country. Are there any hydrological implications, I mean on discharge values whether the predicted temperature was 28 $^{\circ}$C while the observations was 27$^{\circ}$ C?

We agree that the temperature variability in the tropic region is low. However, The VIC model was run for the entire hydrological basins in this region (as mentioned in lines 115-116), where the sources of all major rivers are from the Tibetan Plateau. Consequently, the temperature affects the melting and subsequently the hydrological stream flow.

We will add a sub-basins map (the analysis domain for both climate and hydrology) , see example figure below, that is larger than the MSEA area.

[Figure]

Figure: Subbasins domain map.

160 The fact that the observation driven data set, APHRODITE, "shows 160 a higher skill magnitude compared with the evaluation against WFDE5, especially during the rainy season" might be due to biases in the modeling driven WFDE5 (?).

You make a good point. Generally, the rainfall range (in terms of intensity and frequency) in the tropic region is large. This causes prediction difficulty. The fact that the evaluation with APHRODITE shows a higher skill compared to the evaluation with WFDE5 might be the result that WFDE5 is a model based (though as a reanalysis heavily constrained by data) dataset, and its limited resolution may cause poorer representation of especially rainfall extremes. The bias-correction should reduce some of these errors, but apparently not all. This could be the reason for the slightly larger error in the WFDE5 model. Even though the evaluation with APHRODITE shows a higher skill than with WFDE5, the difference is small.

Figure 7 it appears to me that runoff prediction is better than discharge but the statement of line 217 concluded the opposite.

[Figure]

According to the figure you referred to, the discharge is orange line and runoff is blue line, both representing the correlation coefficient. The discharge and runoff show a similar trend, but it can be seen that the correlation coefficient of discharge shows a little higher score compared to runoff: the orange lines are always above the respective blue lines. Therefore, we conclude that the discharge prediction is higher than runoff.

The problem with this analysis against discharge observation is related to the fact that it includes the uncertainties of the VIC model. Besides all the errors due to the forecasts of the mode, the WFDE5 initialization, etc, how much of the variability can be attributed to the hydrological model itself?

This is a good point and an omission in the paper. It is difficult to define the variability of the VIC model itself because the VIC is data-driven based model. Input data is the main component that give result in variability. We evaluated the correlation coefficient of discharge between real observations (from gauging stations) and WFDE5-driven simulations. You will find the figure below and we will add the figure to the supplement, or combine it with figure 12. The result presents a good correlation coefficient for large parts of the year, so we expect the role of VIC model variability is small in the skill assessments of SEAS5.

Figure S: Monthly correlation coefficient R (p< 0.05) for water discharge generated from VIC model driven by WFDE5 hindcast (reference simulation) against the observation at gauging stations.

---

## Author Comment (AC2)

Dear reviewer,

On behalf of my co-authors, we appreciate for your attention in our paper and the valuable suggestions that very helpful to enhance our manuscript. You will find below the answer according to reviewers' comments.

**Reviewer's specific comments:**

There are several parameters that may need to be calibrated in VIC, but I didn't found descriptions of parameter calibration for the VIC hydrological model. Did the authors just use the default parameters in VIC model?

This is a good point. We used the default parameters for the VIC model. Nevertheless, also in response to a reviewer #1 comment, we evaluated the correlation coefficient of monthly discharge between gauging stations and WFDE5-driven simulation. You will find the figure below and we will add the figure to the supplement, or combine it with figure 12. Generally, the result presents a good correlation coefficient.

[Figure]

Figure S: Monthly correlation coefficient R (p< 0.05) for water discharge generated from VIC model driven by WFDE5 hindcast (reference simulation) against the observation at gauging stations.

The authors sometimes use the term "subseasonal" to describe SEAS5 forecasts, e.g. "ECMWF system 5 (SEAS5) sub–seasonal–to–seasonal (S2S) forecasts". However, from my point of view, SEAS5 mainly provides monthly forecasts and is a seasonal forecast product. Subseasonal forecasts usually refer to daily weather forecasts with lead times between 15-60 days. I suggest the authors to check the use of term "subseasonal" in the manuscript.

Thank you for the comment. We agree that you state the right point. We will substitute and emphasize in our manuscript that we use the seasonal forecast instead of the sub-seasonal forecast.

**Reviewer's Language and technical issues:**

There are too many lines in Fig. 3 and Fig. 5, which makes the figure difficult to understand. The authors may consider improve those figures.

Thank you for the suggestion, We understand that there are many lines in these graphs. Our aim is to present the SESS5 skill of each month from lead-0 to lead-2. We used different colours for each initialization month. However, the forecast skills are different for each initial month and lead times, resulting in these complicated graphs. The fact that the different (black) lines are hard to discern from each other, basically is a good thing: it means that skill, for e.g. temperature, does not decrease much between the first 3 lead months.

We will consider replacing figures 3 and 5 by figures similar to figure 7, so omitting the colored lines.

The language needs to be thoroughly improved. For example, Line 205, "to forecasts" should be "forecast"; Line 227, "skilful" should be "skill" or "skills" in my opinion.

Thank you for the language suggestions. We will revise the language used in our manuscript, assisted by a native speaker.

---

## Author Comment (AC3)

Dear reviewer,

On behalf of my co-authors, we appreciate your attention in our paper and the valuable suggestions that are helpful to enhance our manuscript. You will find below our answer to the reviewers' comments.

**Reviewer's comments:**

This manuscript described an interesting study on evaluation of the forecast skill of SEAS5 in MSEA. Authors concluded that SEAS5 has high forecast skills during the pre-monsoon (April–May) and post-monsoon (October–November), while poor skill is observed during the rainy monsoon season. The paper was written in good style and logical lines. Please see my following comments:

Detailed information of ten streamflow gauging stations should be listed (Lines 103-105). Which basin or sub-basin are these stations located, what are their relationships in terms of upstream and downstream? Figure1, A undelay basin map may be better compared to the country map, same as following figures.

Thank you for the comment. We will add the basin map to Figure 1 and add further station details in a table in the appendix.

[Figure]

| Sub-region | Latitude | Longitude |
|---|---|---|
| North | 23.5° N - 29.0° N | 92.5° E – 100.0° E |
| East | 8.0° N - 24.0° N | 103.5° E – 110.0° E |
| South | 1.0° N - 6.5° N | 98.5° E – 105.5° E |

Please add data availability and data source contents (link for SEAS5, WFDE5, APHRODITE).

We included the references of the driver data in the main manuscript. We will add a data description table (with the data sources) in the Supplement.

Table S2 Data description. Please note, that all data sets have the same native resolution (0.5°)

| Data | Version | Acronym | Time period | Data Sources |
|------|---------|---------|-------------|--------------|
| WATCH Forcing Data | ERA5 | WFDE5 | 1983 - 2014 | https://cds.climate.copernicus.eu |
| ECMWF ensemble forecast | System 5 | SEAS5 | 1985 - 2014 | https://www.ecmwf.int |
| Temperature APHRODITE | 1808 | | 1985 - 2014 | Asian Precipitation-Highly-Resolved Observational Data Integration Towards Evaluation of Extreme Events |
| Precipitation APHRODITE | 1101 | | 1985 - 1997 | |
| Precipitation APHRODITE | 1901 | | 1998 - 2014 | http://aphrodite.st.hirosaki-u.ac.jp/index.html |

As pointed out by other reviewers, the paper is unnecessarily too long because of displaying too many figures and results that can be moved to the supplementary materials. Please consider shorten the paper.

Thank you for the suggestion. We will improve some figures, others will be moved to the supplement. For example, Figures 4 and 6 shows the skill of WFDE5 and APHRODITE and these two are similar; therefore, we will keep one of these in the main manuscript.

Fig. 3, 5, Each colored line follows the skill of a single forecast. Then which color represent which single forecast? The meaning of legend "Lead m 0/1/2" should be explained.

This is a good point. We realized that Figures 3 and 5 are complicated to understand. We will replace figures 3 and 5 by figures similar to figure 7, and omitting the colored lines.

Figure 4,6,8,9 Please consider transfer the figures to seasonal scale (MAM, JJA, SON), which is consistent with later description. There are too many figures in the main text which lead readers confusing. Fig. 13-15 can also be concise, use supplementary to display the repetitive and similar information.

Thank you for the suggestion. We first analyzed the monthly skill for each lead month because our intention is to analyze at highest temporal resolution rather than aggregated seasonal (as mentioned in lines 125-126). This because there can be sharp transitions in skill/discharge

dynamics that do not necessarily align with default meteorological seasons. For the anomalous year analysis, we decided to aggregate the data to the seasonal scale to study the ENSO events. We agree that there are too many figures in our manuscript. We will improve by moving some figures to the supplement.

For hydrological simulation by VIC, the parameter calibration and model validation processes should be clarified. The influencing factors on stream flow should be discussed, like land use change, dam construction, et al. How these factors influencing the forecast skills of SEAS5, can be discussed. These required a basin-to-basin analysis in MSEA, please authors consider compare the basin variation characteristics, rather than the sub-region analysis in current version.

You made a good point. We agree that there are factors that related to streamflow, such as land use. However, our main focus is climate forecasting and subsequently use the results for streamflow forecasting. We studied different sub-regions because the climate factors influence gridded streamflow differently among these sub-regions. However, when discussing results at station level, a discussion of upstream basin characteristics is definitely relevant, so we will add a few lines accordingly. It is correct that a basin analysis should explain more about the streamflow characteristics, we will pay more explicit attention to this in future research.